# Establishing a High-Throughput Locomotion Tracking Method for Multiple Biological Assessments in *Tetrahymena*

**DOI:** 10.3390/cells11152326

**Published:** 2022-07-28

**Authors:** Michael Edbert Suryanto, Ross D. Vasquez, Marri Jmelou M. Roldan, Kelvin H. -C. Chen, Jong-Chin Huang, Chung-Der Hsiao, Che-Chia Tsao

**Affiliations:** 1Department of Chemistry, Chung Yuan Christian University, Chung-Li 320314, Taiwan; michael.edbert93@gmail.com; 2Department of Bioscience Technology, Chung Yuan Christian University, Chung-Li 320314, Taiwan; 3Department of Pharmacy, Faculty of Pharmacy, University of Santo Tomas, Manila 1015, Philippines; dvasquez@ust.edu.ph; 4Research Center for the Natural and Applied Sciences, University of Santo Tomas, Manila 1015, Philippines; 5The Graduate School, University of Santo Tomas, Manila 1015, Philippines; 6Faculty of Pharmacy, University of Santo Tomas, Espana Blvd., Manila 1015, Philippines; mmroldan@ust.edu.ph; 7Department of Applied Chemistry, National Pingtung University, Pingtung 900391, Taiwan; kelvin@mail.nptu.edu.tw (K.H.-C.C.); hjc@mail.nptu.edu.tw (J.-C.H.); 8Center of Nanotechnology, Chung Yuan Christian University, Chung-Li 320314, Taiwan; 9Research Center of Aquatic Toxicology and Pharmacology, Chung Yuan Christian University, Chung-Li 320314, Taiwan; 10Department of Biological Sciences and Technology, National University of Tainan, Tainan 70005, Taiwan

**Keywords:** *Tetrahymena*, protozoa, TRex, locomotion, toxicity, complexity reduction

## Abstract

Protozoa are eukaryotic, unicellular microorganisms that have an important ecological role, are easy to handle, and grow rapidly, which makes them suitable for ecotoxicity assessment. Previous methods for locomotion tracking in protozoa are largely based on software with the drawback of high cost and/or low operation throughput. This study aimed to develop an automated pipeline to measure the locomotion activity of the ciliated protozoan *Tetrahymena thermophila* using a machine learning-based software, TRex, to conduct tracking. Behavioral endpoints, including the total distance, velocity, burst movement, angular velocity, meandering, and rotation movement, were derived from the coordinates of individual cells. To validate the utility, we measured the locomotor activity in either the knockout mutant of the dynein subunit DYH7 or under starvation. Significant reduction of locomotion and alteration of behavior was detected in either the dynein mutant or in the starvation condition. We also analyzed how *Tetrahymena* locomotion was affected by the exposure to copper sulfate and showed that our method indeed can be used to conduct a toxicity assessment in a high-throughput manner. Finally, we performed a principal component analysis and hierarchy clustering to demonstrate that our analysis could potentially differentiate altered behaviors affected by different factors. Taken together, this study offers a robust methodology for *Tetrahymena* locomotion tracking in a high-throughput manner for the first time.

## 1. Introduction

Aquatic protozoa are unicellular, phagotrophic organisms and major consumers of microbes in the environment [1]. Unlike bacteria or algae, protozoa do not have a cell wall and can respond quickly to external stimuli [2], which are useful when studying the effects of pollution on the microbenthic fauna [3]. Their microscopic size and short cell cycles make it easy for scientists to analyze the effects of pollutants on cell viability [4].

*Tetrahymena* is a ciliated protozoan that inhabits streams, lakes, and ponds in a range of climates [5]. This ciliate, along with other members of the ciliate family, is vital for energy transfer within the microbial loop and serves as an early warning indicator of negative environmental impacts through its activity [6]. Selected species in the *Tetrahymena* genus can be easily cultured in the axenic medium with a short generation time [7] and have become important model systems for biological research. As an eukaryotic cell with a dozen micrometers in any dimension, *Tetrahymena* exhibits a complexity in the subcellular organization comparable to human cells, making them a good alternative to human tissues [8]. Asexually reproduced *Tetrahymena pyriformis* is frequently used as one of the representative aquatic environmental microorganisms to investigate the potential toxicity of various xenobiotics [6,9,10]. Meanwhile, with the molecular toolkit available, closely related *Tetrahymena thermophila* is subjected to genetic manipulation and systems biological analysis, offering a potential to functionally dissect the underlying responsive pathway [11,12,13].

Different chemicals, including pharmaceuticals, heavy metals, pesticides, and organic compounds, have been applied to *Tetrahymena* for in vitro toxicological assessment [14,15,16,17,18]. In these assays, the inhibition of cell proliferation and/or viability is a common endpoint to indicate gross physiological impairment [19]. Meanwhile, *Tetrahymena* cells lost their overall appearance by changing from the pear-like to round shape, decreasing in size, and showing a bulged cell membrane after the exposure to toxicants [20,21,22]. Changes of the organelle have also been documented, such as the compromised morphology or integrity in the macronucleus, mitochondria, peroxisome, cytoplasmic granule, and contractile vacuole [23]. In addition, physiological responses of *Tetrahymena* can be useful for assessing the effect of toxicants. Physiological states and environmental factors, such as temperature, pH, and oxygen level, affect food vacuole formation [24,25]. The contractile vacuole, which is responsible for maintaining the osmotic balance of the cells, was shown to be sensitive to changes in the environment [26]. Such analyses can be informative to determine the cellular function perturbed by the reagent; however, laborious handling to execute these detailed investigations and, sometimes, the demand for expertise preclude them from being adopted as a rapid, robust, and routine method.

Studies on behavioral parameters offer advantages in toxicology assessments. Comparative to survival bioassays, behavioral effects are considered to be the most sensitive indicators of environmental changes because they occur quickly and can be detected at low doses and in non-invasive ways on living organisms [27]. In *Tetrahymena*, behavior studies mainly focus on phagocytosis activity [28], chemotaxis response [29], and cell motility [30]. Among them, cell locomotion is easy to monitor to directly manifest the ciliary motility. Previous studies demonstrated the correlation between cell movement and exposure to carbon nanotubes, metals, and pesticides in ciliates [31,32,33,34]. For example, a high concentration of lanthanum ions causes a decrease in cell motility and the rocking movement [35]. Interestingly, researchers developed an in vitro computerized test for chemical toxicity using Tetrahymena swimming patterns as an alternative to the Draize rabbit eye irritation test. The scoring of how *Tetrahymena* change their swimming patterns in a hostile chemical environment yields results that compare favorably to the in vivo animal test with equal or better detection limits [36], which argues the importance of developing a pipeline that allows to analyze *Tetrahymena* locomotion rapidly, quantitatively, and comprehensively.

Once the improvement of behavioral studies in *Tetrahymena* allows for collecting large and detailed datasets, it will become a massive task to analyze them for the complexity, nonlinear dependency, and uncertainty associated with these diverse, complex, and often high-dimensional datasets using traditional statistic methods [37]. Therefore, dimensionality reduction is needed to reduce the high-dimensional data to low-dimensional data to achieve representation of some meaningful properties from the original data, ideally close to their intrinsic dimension [38]. Methods such as principal component analysis (PCA), hierarchical clustering, and self-organizing maps facilitate a great reduction in dimensionality (complexity) of data and models. Typically, the results are generated into plots that represent the behavior of the samples (score plot) or variables (loading plot) in two or three dimensions [39].

A previous study to analyze protozoan movement was mainly based on image analysis and quantifying their motility in terms of the swimming speed (Table A1 in Appendix B). In general, these analyses require several manual steps, come with low-throughput performance, and have limited measurement endpoints preset by the software utilized. To further advance the employment of *Tetrahymena* as a simple and practical model for various biological studies, here we developed a high-throughput method to track cell locomotion using a software with a computer vision approach. Multiple locomotor endpoints could be assessed in the workflow. We demonstrated that this method was effective to analyze *Tetrahymena* cells of different motility caused by genetic background, nutrition state, or copper exposure. We also applied an unsupervised intuitive method that assigns variable groups of *Tetrahymena* locomotion endpoints to smaller variables so that the behavioral effect can be more readily visualized and understood.

## 2. Materials and Methods

### 2.1. Tetrahymena Strains and Culture

The *Tetrahymena thermophila* CU428 wild-type strain was received from Dr. Meng-Chao Yao’s laboratory (Academia Sinica, Taipei, Taiwan). The DYH7neo3 mutant was purchased from *Tetrahymena* Stock Center at Cornell University (Ithaca, New York, NY, USA). The cultures were maintained at 27–32 °C in the SPP medium [40] containing 1% proteose peptone, 0.1% yeast extract, 0.2% dextrose, and 0.003% sequestrene. Cells were grown in a glass beaker (PYREX^®^ 3000 mL) (Corning Inc., New York, NY, USA) as the original stock. Horizontal shaking at 150 rpm was applied for aeration. The cell was grown to the middle log phase (1–2 × 10^5^ cells/mL) for the cell locomotion tracking test.

### 2.2. Microscope Setup for Tetrahymena Observation

From the original stock, 100 μL of cells were dispersed into six-well depression concave slides. *T. thermophila* movement was observed using the upright microscope (ex20, SOPTOP, Taipei, Taiwan) with the 4× plan objective. A high-resolution 4K CCD (XP4K8MA, ToupTek, Zhejiang, China) was mounted onto the microscope to record the *T. thermophila* locomotion at the resolution of 3840 × 2160. The video was recorded for 10 s at 30 frames per second (fps) and saved in the .mp4 format. The recording setup was established based on a previous publication [41].

### 2.3. Locomotion Tracking Using TRex

The video of *T. thermophila*’s movement was analyzed by TRex (https://trex.run/, accessed on 28 May 2021), a software (version 1.1.8_1, Tristan Walter, Konstanz, Germany) designed to track and identify individual entities based on computer vision and machine learning [42]. The source-code was downloaded from https://github.com/mooch443/trex (accessed on 28 May 2021). The workload of this software is divided into two packages: TGrabs and TRex. The recorded video was opened by TGrabs with the default setting, whereas the threshold value was lowered to 15 due to the small foreground objects (blobs) and noise that were detected in the video. TGrabs enabled us to estimate an individual object’s position, posture, and visual field, and then convert the existing videos into a custom non-proprietary video (PV) format. Then, the track-converted videos in the PV format were analyzed by TRex with settings of blob size ranging from 0.01 to 0.1 pixels, threshold of 15, and enabled automatic recognition and posture calculation. The specific tracking parameters, such as position of centroid in XY pixel coordinates, speed, and trajectories, from each individual object frame-by-frame were exported and stored in separate numpy array gzip compression data (npz) files. To simplify the output data analysis, we changed the output format from npz to a comma-separated values (csv) file that can be easily accessed by Microsoft Excel. Detailed steps for the workflow and software settings are described in the Appendix A, “Establishing a Simple and High-Throughput Toxicity Assessment in *Tetrahymena* (SOP)”.

### 2.4. XY Trajectory Data Extraction and Locomotor Endpoint Calculation

The trajectory data saved in .csv files were opened by Microsoft Excel. Since TRex saves the output XY coordinates of each individual object to individual files separately, a macro script based on Microsoft Excel Visual Basic for Applications (VBA) was used to automate tasks by which multiple spreadsheets from individual files were combined into a single spreadsheet. The locomotor endpoints based on the XY trajectory data were calculated using a spreadsheet in the combined file according to the locomotor endpoints listed in Table 1. The trajectory plot of *T. thermophila* locomotion was generated by using the software OriginPro, version 2019b 64-bit. The X and Y coordinates of each cell were input to the workbook and plotted as a line. The trajectory line of *T. thermophila* locomotion was displayed for a total of 10 s. Detailed steps for the workflow and software settings are described in the Appendix A, “Establishing a Simple and High-Throughput Toxicity Assessment in *Tetrahymena* (SOP)”.

### 2.5. Starvation and High-Nutrient Medium Conditions

*T. thermophila* locomotion was evaluated in the medium with two nutrition conditions. For the high-nutrient medium, the axenic proteose peptone-based SPP [40] was used. For the starvation condition, CU428 cells were grown in 100 mL of double-distilled water (ddH_2_O) supplemented with 5 grains of wheat seeds. The temperature was sustained at 27–32 °C for both conditions. On the following day, a 100 µL sample from each group was taken to observe under the microscope, and the swimming movement was recorded for 10 s at 30 fps.

### 2.6. Copper Exposure Treatment

Copper sulfate (CuSO_4_, from Shanghai Macklin Biochemical Co., Ltd., Shanghai, China) was prepared as stock solution in ddH_2_O. CuSO_4_ was selected as a representative of heavy metal toxicants because it is widely known to have profound effects on ciliated protozoans [43]. Five different concentrations (1; 10; 100; 1000; and 10,000 µM) were prepared to conduct the toxicity test. In total, 100 µL of *T. thermophila* was placed on a six-well depression concave slide that had circular raised ring wells with a diameter of 13 mm and depth of approximately 0.5 mm. A total of 100 µL of CuSO_4_ was immersed by mixing it with the cells on the slide until the desired working concentrations were reached (0.5, 5, 50, 500, and 5000 µM) and were exposed for 30 min at room temperature. The concentrations were set based upon the maximum concentration that does not affect the growth rate of the cells [44]. *T. thermophila*’s swimming movement was recorded for 10 s at 30 fps.

### 2.7. Calculation of EC_50_

EC_50_ is the effective concentration of the toxicant that causes a 50% reduction in cell locomotion after a specified exposure time. The swimming inhibition of *T. thermophila* cells after 30 min exposure to CuSO_4_ was evaluated. The average speed endpoint data were normalized to percentage values with respect to the zero copper control set, and the concentrations were transformed to log10 values. The concentration-effect curve, fitted by the log inhibitor versus the normalized response model, was constructed, and the EC_50_ value with a 95% confidence interval was calculated based on log concentrations using the software package GraphPad Prism (version 8.0.2., GraphPad Software, Inc.: San Diego, CA, USA).

### 2.8. Statistical Analysis

Statistical tests were performed and graphic plots were made using GraphPad Prism software (version 8.0.2., GraphPad Software, Inc.: San Diego, CA, USA). The nutritional condition and mutant strain comparison tests were carried out with a non-parametric Mann–Whitney analysis since the recorded data were not normally distributed. For the copper exposure test, the data were analyzed using a one-way ANOVA with a non-parametric Kruskal–Wallis multiple comparisons test. All the presented data are shown as the median and interquartile range. Data were considered statistically significantly different when * *p* value < 0.05, ** *p* value < 0.01, *** *p* value < 0.001, and **** *p* value < 0.0001.

### 2.9. Principal Component Analysis and Hierarchical Clustering

Using the ClustVis web tool (https://biit.cs.ut.ee/clustvis/, accessed on 13 March 2022) (ClustVis version 2018, University of Tartu, Tartu, Estonia) [45], the principal component analysis (PCA) and hierarchical clustering analysis were conducted. The data matrix was generated by using six different locomotor activity endpoints (total distance, average speed, meandering, angular velocity, burst movement, and rotation movement). An Excel spreadsheet was used to summarize every average value of each locomotor activity endpoint. The spreadsheet then was saved as a csv and the file was uploaded to ClustVis. For the preprocessing options, the default setting with no transformation logarithmic function was applied. Unit-variance scaling was used for each row to ensure that all variables were treated equally. To calculate principal components, singular value decomposition was selected since there were no missing values in the dataset. For the hierarchical clustering, the heatmap was generated by calculating all pairwise distances. As each step proceeded, the objects with the smallest distance were merged. The correlation distance and average linkage were used in clustering for both rows and columns. Afterward, the PCA plot and heatmap results were exported and saved to the system.

## 3. Results

### 3.1. Overview of the Experimental Design and Workflow

One drawback to employing protozoan behavior as a toxicological testing endpoint is the relatively low throughput needed to analyze the result. Usually, this requires manually selecting the object to be analyzed from the recording video. Meanwhile, some software allows automatic cell tracking, but the specifically designed package or plug-in focuses on selected default parameters with limited flexibility to modify them. With recent advances and the public accessibility of artificial intelligence-based computing, we sought to establish a high-throughput and customable method to analyze protozoan locomotion. The *T. thermophila* movement was recorded in a video and analyzed using TRex, a software designed to track and identify individual entities based on computer vision and machine learning [42]. This tool could automatically count individual *T. thermophila* cells, analyze their movement, and store the XY trajectory position of each cell from each frame. The output result for each cell that was detected during the analysis was saved into individual files separately (Figure 1A). However, the subsequent analysis step would be quite difficult to proceed with since hundreds of cells were tracked in the video and hundreds of files would need to be cross-referenced. Therefore, we used the VBA to compile all the data from each individual *T. thermophila* into a single spreadsheet file (Figure 1B) to facilitate further analysis and calculation. Finally, the footage of the *T. thermophila* movement was visualized using Origin 2018 (version 9.1, OriginLab, Northampton, MA, USA) a graphing and analysis software. The *T. thermophila* trajectory plot (Figure 1C), which was created based on X and Y coordinates from each cell per frame of the duration of the video, provides a holistic glimpse of the tracking result.

### 3.2. Recording Setup and Multiple Endpoints’ Calculation

To test the recording setup and analyzing workflow, we firstly performed the locomotion analysis on the wild-type *T. thermophila* CU428 strain without any special treatment. All videos were captured by a high-resolution CCD mounted onto an upright microscope with a 4× objective lens [41]. Since a good video recording process is crucial to ensure optimal tracking performance, we found that videos with a resolution of 3840 × 2160 pixels and a frame rate of 30 frames per second could provide an optimal recording and achieve high contrast suitable for TRex tracking and analysis.

After the tracking results from TRex were collected and composed into a single spreadsheet file, we used it to calculate multiple locomotion endpoints based on movement trajectories. The trajectories were quantified and interpreted through six locomotor endpoints, including total distance, velocity (average speed), burst movement, angular velocity, meandering, and rotation movement (Table 1). Total distance is the total length of the path traveled by the cells. Average velocity (speed) is the change of a cell’s position in a particular direction divided by the time it took to travel from the initial position to final position. Burst movement is the spontaneous movement of cells, which is indicated by the high swimming speed (above the upper quartile of the average speed). Angular velocity is a measurement of the rate of change of angular position from a cell over the period. Meandering is the measurement of the change in direction of a body point relative to the distance moved. Lastly, rotation movement is a total circular motion, either clockwise or counterclockwise. All quantifications were calculated in Microsoft Excel and the formulations that were used in this study are listed in the Appendix A.

Based on this setting and analyzing workflow, we successfully obtained six locomotor activity endpoints from hundreds of untreated wild-type cells. We found the total distance traveled within 10 s was 1.437 ± 0.420 mm (Figure 2A), the average speed was 0.179 ± 0.031 mm/s (Figure 2B), the total burst count was 46.56 ± 2.345 times (Figure 2C), average angular velocity was 136.9 ± 1.995 (Figure 2D), meandering movement was 2.907 ± 0.052 (Figure 2E), and total rotation movement count was 6.707 ± 0.151 times (Figure 2F).

### 3.3. Validation Using a Motility Mutant Strain

To further evaluate whether our tracking analysis can effectively differentiate the behavioral difference between two testing samples, we took advantage of the fact that *T. thermophila* is an established and resourceful model organism for biological research. We analyzed the locomotion between the wild-type CU428 strain and a motility mutant DYH7neo3 [46]. The DYH7neo3 strain was created by knocking out the *T. thermophila DHY7* gene, which encodes the heavy chain of the ciliary dynein arm complex that generates the ciliary motility to propel cell locomotion. With the defective dynein complex, the cilia of this mutant were observed to entangle and obstruct neighboring cilia with a less coordinated, swiveling movement, which causes the DYH7neo3 mutant cell to display altered movement with a reduced swim speed [46] (Appendix B). In line with those previous findings, our present study also revealed a significant reduction in the total distance traveled (0.859 ± 0.464 for mutant vs. 1.434 ± 0.396 for control), average speed (0.102 ± 0.059 for mutant vs. 0.180 ± 0.027 for control), and burst movement (26.72 ± 3.606 for mutant vs. 45.57 ± 2.965 for control) for the DYH7neo3 mutant with a *p* value < 0.0001 (Figure 3A–C). In addition, we also checked other locomotor endpoints that were not documented in the previous study, including the average angular velocity, meandering, and rotation movement. Compared to the wild-type CU428, a significant difference was displayed in the increased average angular velocity (146.8 ± 2.560 for mutant vs. 135.6 ± 2.826 for control, *p* value < 0.05) (Figure 3D), increased meandering movement (7.785 ± 0.559 for mutant vs. 2.877 ± 0.062 for control, *p* value < 0.0001) (Figure 3E), and increased rotation movement count (6.736 ± 0.165 for mutant vs. 5.628 ± 0.166 for control, *p* value < 0.0001) (Figure 3F). Overall, in this proof-of-concept validation, our tracking analyses effectively demonstrate that the mutant strain has lower swimming activity compared to the wild-type as we expected. Furthermore, the behavioral difference between the two strains due to the dynein motor protein mutation could be manifested in six locomotor endpoints using our analysis.

### 3.4. Analysis of T. thermophila Locomotor Activities in Different Nutrient Conditions

We observed and noticed that *T. thermophila* exhibited different motility when they were maintained in the different mediums supplied with low or high nutrients. We decided to use our locomotion tracking to quantitatively analyze the difference. For the high-nutrient medium, we used the axenic proteose peptone-based SPP medium, which is commonly used for growing *T. thermophila* in the laboratory. For the low-nutrient (“starvation”) condition, we grew the *T. thermophila* in distilled water supplemented with wheat seeds. As can be expected, *T. thermophila* in the low-nutrient medium displayed lower locomotion activity compared to the high-nutrient group (Appendix B). We found that the total distance traveled (0.991 ± 0.047 mm for starvation vs. 2.118 ± 0.092 mm for rich medium) and average speed (0.156 ± 0.007 mm/s for starvation vs. 0.213 ± 0.009 mm/s for rich medium) in the low-nutrient group were significantly reduced (with *p* value < 0.0001, Figure 4A–C). The meandering movement was also affected by the lack of nutrient treatment. In the low-nutrient group, it was significantly higher (6.757 ± 0.339) compared to the high-nutrient group (1.124 ± 0.055) with a *p* value < 0.0001 (Figure 4D). We also analyzed their burst and rotation movements (Figure 4E,F) and found that both endpoints in the low-nutrient condition were significantly lower compared to the PP-based media group (*p* value < 0.0001).

### 3.5. Analysis of T. thermophila Locomotor Activities in the Exposure to Copper Ions

Further assessment by observation and evaluation of *T. thermophila* as a promising model organism for toxicological testing was carried out by comparing their locomotion activity when exposed to multiple concentrations of copper sulfate (CuSO_4_), which were 0.5; 5; 50; 500; and 5000 μM. The chemical was applied for 30 min and the locomotion activity of *T. thermophila* was recorded for 10 s and analyzed. Despite the short span of exposure time, *T. thermophila* already displayed marked behavioral changes based on their total distance traveled and average speed (Appendix B). The reduction in the total distance traveled was shown from the lowest to the highest concentration of CuSO_4_ (Figure 5A). Meanwhile, the reduction in the average speed started to be observed from 5 μM CuSO_4_ exposure. Compared to the control, a significantly lowered average speed in the 5, 50, and 500 μM CuSO_4_ treatment was observed (Figure 5B). The burst movement was also significantly reduced (Figure 5C). The angular velocity, however, was not affected by CuSO_4_ treatments, except at the highest concentrations (500 and 5000 μM, Figure 5D). On the other hand, the meandering movement with 5–5000 μM of exposure displayed a significant increase compared to the control (Figure 5E). The CuSO_4_ treatment also caused more rotation movement when cells were exposed to 5 μM or higher (Figure 5F). We noticed that all the locomotor endpoints in the highest concentration group (5000 μM) were strikingly deviated from those obtained in other treatments or the control group. This could be due to the toxicity effect from the high concentration of CuSO_4_ that was too severe for *T. thermophila* and led to high mortality. Overall, these results indicate that CuSO_4_ indeed induced a stress in *T. thermophila* cells and affected their motility behavior. Our analysis could serve as a quantitative tool to assay the effect. Based on the average speed, we also calculated the EC_50_ of CuSO_4_, which caused 50% swimming inhibition of *T. thermophila*’s overall locomotion at the concentration of 154.0 µM (Figure A2).

### 3.6. Reducing the Data Complexity Using PCA and Clustering

To evaluate and compare the behavioral endpoint similarities or differences in *T. thermophila* triggered by different treatment conditions, we performed PCA and hierarchical clustering based on the data retrieved from the locomotor activity tests. Using the hierarchical clustering algorithm, three major clusters were identified (Figure 6A). In the first cluster, representing the mildly affected condition, the control, CuSO_4_ 0.5 µM, and the low nutrient group were grouped in a single major cluster. The second cluster consists of the copper exposure treatment with different concentrations (5, 50, and 500 µM). The third cluster, which was the most distinct group, contains the mutant strain DYH7neo3 and the highest concentration of CuSO_4_ exposure (5000 µM). This result indicated that the effects of the low-nutrient and 0.5 µM CuSO_4_ exposure treatment were less severe than those in the other treatment groups since they belong in the same group together with the control. Meanwhile, the mutant and 5000 µM copper group displayed the most marked change compared to the control, as revealed by its distant position from the other two clusters. In addition, a better comprehension of different *T. thermophila* locomotor activities among treatment groups could be obtained based on the PCA plot (Figure 6B). The low-nutrient group displayed a similar locomotor activity pattern with the copper group and were located in the same region on the plot. In the meantime, the DYH7neo3 mutant displayed a different behavioral pattern compared to the other groups. The 5000 µM copper group, which was severely intoxicated, was also positioned as remotely distant from the control. Therefore, clustering hierarchy and PCA plot proximity based on our locomotor endpoint data could reflect the locomotion difference among *T. thermophila* cells, and such difference could indeed be affected by different contributing factors such as a genetic defect, nutritional state, or ion exposure.

## 4. Discussion

### 4.1. Tetrahymena thermophila Locomotion Is a Sensitive Biomarker for Toxicity Assessment

Behavioral activities, which link biochemical processes on the organism level to their surrounding ecological consequences, can be observed at short response times in real time and on a repeated basis, allowing for high-throughput analysis [27]. In prior studies, *Tetrahymena* locomotion activity was shown to be disrupted under exposure to copper [47], morphine [48] or single-walled carbon nanotubes [31]. Another study investigated the change in swimming speed with the effect of different glucose concentrations [49]. Since *Tetrahymena* lives in an environment where swimming is required, locomotor activities are important and valuable indicators of cell physiology.

In this study, we analyzed the motility based on the movement trajectory of *T. thermophila* where the XY coordinates of hundreds of cells were recorded. Based on those coordinates, we calculated six locomotor endpoints to evaluate the locomotion change more comprehensively (Table 1). Total distance and average speed are important endpoints that have been commonly used in locomotion studies [50]. The total distance traveled by each cell was determined by calculating the distance between the location from each frame of each cell. The average swimming speed was determined through dividing the total distance traveled by the time the organism spent in the video. *Tetrahymena* cells typically move with a burst of forward movement, but occasionally they tumble [51], especially when in the presence of a repellent or particular substance. We, therefore, also evaluated the burst movement count of *T. thermophila* throughout the time period. The angular velocity, which refers to the speed of a rigid body rotating with respect to its center of rotation, means how fast the cell makes a rotating movement. In addition, we also calculated the total rotation count. Lastly, we evaluated the meandering movement, which is a result of the interaction between the animal/cell and water and which has been applied in other animal models such as worm [52], fish [53], and snail [54]. Meanders change position by eroding sideways, and meandering is calculated based on the body angle profiles [52]. To summarize, unlike previous software-aided analyses which usually only calculate limited preset endpoints, our method offers an opportunity to assess multiple aspects of cell locomotion in a single experiment (Figure 2).

We tested several conditions to verify the performance of our method and to demonstrate its potential in multiple biological assessments. *Tetrahymena* movement is coordinated by the beating of hundreds of cilia that cover its body. The outer and inner arm of axonemal dyneins are responsible for the generation of the force for ciliary movement. A knockout mutant, DYH7neo3, which lacks the inner arm dynein of the heavy chain protein Dyh7p, led to an abnormal formation of the ciliary waveform with asynchrony patterns of beat and erratic bend propagation [46]. Therefore, we used this strain as the proof-of-concept trial in this study, and our results support the previous finding that the swimming velocity was significantly reduced (Figure 3A,B), which validates our method and further provides evidence of the altered movement from this mutant strain.

It is commonly known that a gradual decrease in swimming speed occurs as a result of *Tetrahymena* starvation [55]. Our analysis clearly showed that most of the locomotor activities were significantly reduced in the starvation condition (Figure 4). We reason that this phenomenon might be due to a decrease in the ATP pool in *Tetrahymena* [56], and the lack of nutrients might also cause stress to the cell [57]. Lastly, the copper toxicity of *T. thermophila* was evaluated using our method. Our analysis revealed that the increased concentration of copper gradually caused a severe locomotor disruption (Figure 5). In this study, the EC_50_ was calculated based on the swimming speed of *T. thermophila* cells after 30 min exposure to CuSO_4_. A similar approach was used to obtain the EC_50_ of lofepramine based on the swimming speed of *T. pyriformis* [58]. With simple video recording and a short amount of time, the results of the quantitative measurements were retrieved in our analysis. In a previous study, EC_50_ values that showed viability of *T. thermophila* after 24 h of exposure to CuSO_4_ in different media and vessels varied from 27.753 to 132.2 μM [59]. Another study evaluated the cellular proliferation of *T. thermophila* under copper stress. The inhibition rate (IC_50_) value was displayed at a concentration of 1.80 mM using the decreasing cell number after 6 h of CuSO_4_ exposure [60]. Compared to prior studies, our method may be considered as a rapid alternative to substitute for the traditional EC_50_ measurements that are based on growth or mortality, which are laborious and time-consuming.

Regarding copper toxicity, a previous study demonstrated a trend toward slower moving cells in a higher concentration of heavy metal, CuSO_4_ [47]. As copper concentrations increased, fitness decreased and death rates increased. Naitoh and Eckert (1969) showed that changes in ion permeability affect the direction and rate of ciliary beats, especially calcium and potassium ions, thus changing the membrane potential [61]. Since it is possible for toxicants to inhibit swimming speed by altering ion fluxes, our results suggested that copper ions might have entered the *T. thermophila* cell. We speculate that with the increase in the amount of metal ions in *Tetrahymena* cells, it will lead to cell membrane damage [62] and the membrane will be unable to selectively absorb the extracellular substance resulting in the gradual disappearance of its functionality [63]. This will trigger a change in intracellular function that hinders the cell from obtaining nutrients and a decrease in cell fitness that causes a slower swimming speed [64]. However, the exact mechanism is still to be elucidated.

We applied the mathematical tool that reduces the data dimensionality to facilitate visualization and interpretation of the results from the comparison of various cell conditions. The PCA and clustering analysis based on all six locomotor endpoints showed that DYH7neo3 is located distinctly on the plot as compared to the other cell treatments in this study (Figure 6). Since the DYH7neo3 mutant is a cell with the targeted deletion of a ciliary gene, but otherwise in normal condition, its ciliary beating is affected in a specific manner. On the other hand, in the cell under starvation or exposure to copper ions, the locomotion is more likely to be affected grossly because of the compromised physiological condition. Our PCA analysis clearly demonstrated the difference, and this result argues that our method is sensitive enough to detect and to distinguish the alteration of cell behaviors under different scenarios, which should be instrumental to shed light on differentiating toxicological mechanisms.

### 4.2. The Pros and Cons of the TRex-Based Locomotion Tracking Method

In this study, the *T. thermophila* movement was recorded and the video of locomotion tracking was analyzed using TRex [42], which has high accuracy and rapid performance suitable to detect individual movements of small objects. The TRex was originally used to track fish, fruit flies, termites, and locusts; however, the developer stated that their software is not just limited to such species. Tracking moving objects/animals is possible by TRex as long as the background can be separated from the object. Within the software package, there is a task-specific tool, TGrabs, for converting videos and segmenting them into separated background and foreground objects, which then enables TRex to recognize the moving objects, such as protozoa. Another advantage of using this software is its capability to track up to 1024 individuals, allowing to collect enough data for analyses with decent statistical power. Furthermore, we used the Excel VBA to circumvent the issue of tedious working steps that combine and handle data of individual cells from individual files. With this tool, we can automate Excel tasks and apply them to a large scale of multiple sheets. After that, locomotor endpoints were obtained for evaluation. All the endpoints were calculated in the master spreadsheet using the custom-written formula (Table 1), which offers the flexibility to analyze and expand to other locomotor behaviors as long as they can be defined and derived from the XY coordinates.

There are limitations and precautions that come with adopting our method. First, errors could be reported by TRex as some individual cells might move and swim outside the video frame. When this occurs, identity loss and mislabeling may happen, and thus, some manual checking and correction are still needed to filter out the mistake. We estimated there was an approximate 13–17% error rate from the overall data. Limited selection of threshold parameters for adjustment in TRex is also a disadvantage of this tool. With the ability only to gauge by size and pixel threshold, we did not have many options to adjust the tracking parameters. This limitation brings a potential weakness, which is the inability to distinguish between different protozoan species, other animals, or contaminants that are found in the water used for testing if the sample contains heterogeneous objects. At the current stage, this is not a major problem since most of the ecotoxicological assays are carried out using a single, selected model organism in the laboratory. Eventually, with the promising prospect of the machine learning and artificial intelligence era, hopefully in the future the updated TRex can escalate the scale of its tracking performance to be suitable for composite samples.

In conclusion, the remarkable difference in multiple locomotor behavior endpoints can be observed in *Tetrahymena* using our method. With the ability to observe the locomotor activity alterations in cells treated with different conditions, the excellence of *Tetrahymena* as a simple model organism for biological research can be highlighted. In the future, we would like to perform a large-scale screen of potential pollution chemicals, such as rare earth elements, antidepressants, pesticides, and other freshwater pollutants, that might alter locomotor activity in *Tetrahymena*. Hopefully, by using the high-throughput analysis of the *Tetrahymena* locomotion activity, our established method and simple setup of a behavior assessment will provide a useful approach to further improve the research and investigations in various laboratory research fields.

## Figures and Tables

**Figure 1 cells-11-02326-f001:**
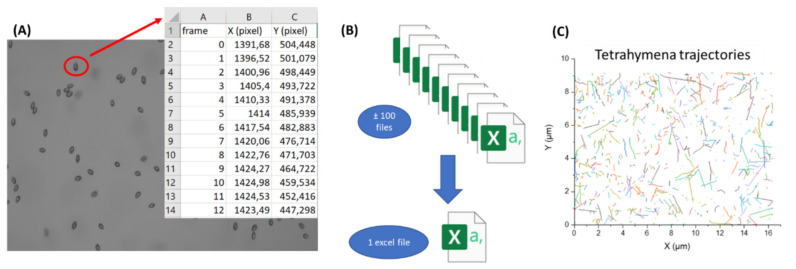
Experimental workflow for *T. thermophila* locomotion tracking and locomotor activity measurement. *T. thermophila* samples were transferred to six-well depression slides that were able to accommodate 100–200 µL samples. Original video was captured for 1 min for each sample by a high-resolution CCD mounted onto upright microscope. Later, this 1 min video was handled by TGrabs tool for object identification. Finally, each individual cell’s locomotor trajectory and XY coordinates were tracked by the TRex tool. (**A**) The output result of XY trajectory from one individual *T. thermophila* cell. (**B**) Combining all the data from separate individual cells into a single spreadsheet file using VBA. (**C**) The trajectory footage of all *T. thermophila* movement captured in the video.

**Figure 2 cells-11-02326-f002:**
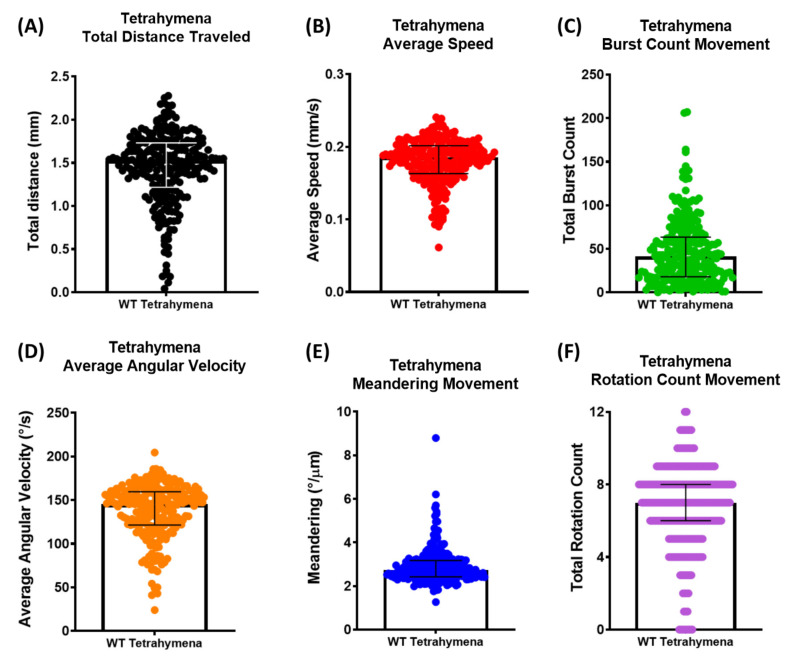
Summary of multiple endpoints of locomotor activity analyzed in *T. thermophila*. (**A**) Total distance traveled (mm), (**B**) average speed (mm/s), (**C**) total burst movement count, (**D**) average angular velocity (°/s), (**E**) meandering (°/µm), and (**F**) total rotation movement count were calculated based on the 10 s video recording. Median and interquartile range are used to express the data (*n* = 249).

**Figure 3 cells-11-02326-f003:**
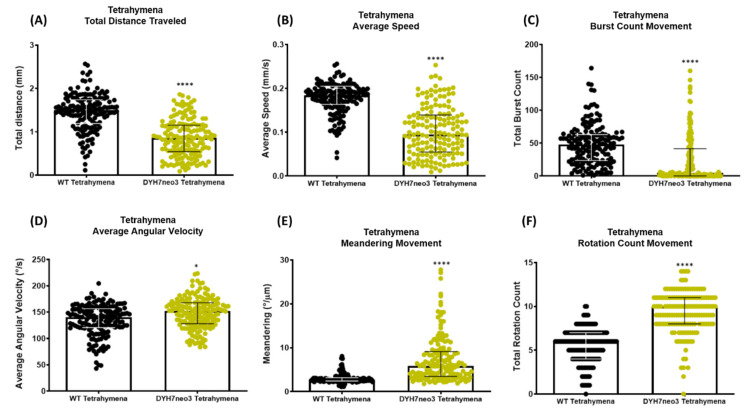
Comparison of locomotor activity between wild-type (CU428) and mutant (DYH7neo3) *T. thermophila* for 10 s. Six locomotor endpoints of (**A**) total distance traveled, (**B**) average speed, (**C**) burst movement, (**D**) average angular velocity, (**E**) meandering, and (**F**) rotation movement were statistically analyzed by Mann–Whitney test (*n* = 158 for each group; * *p* value < 0.05; **** *p* value < 0.0001). Median and interquartile range are used to express the data. Cumulative trajectory paths are presented in Figure A1.

**Figure 4 cells-11-02326-f004:**
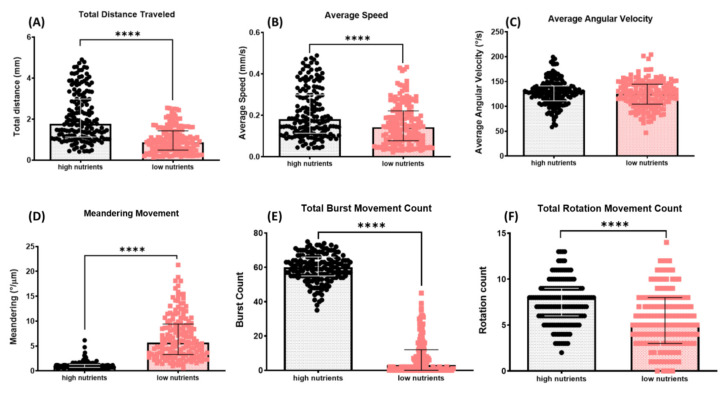
Comparison of *T. thermophila* locomotor activity in high- and low-nutrient media for 10 s. Six locomotor endpoints of (**A**) total distance traveled, (**B**) average speed, (**C**) average angular velocity, (**D**) meandering movement, (**E**) burst movement, and (**F**) rotation movement were statistically analyzed by Mann–Whitney test (*n* = 167 for each group; **** *p* value < 0.0001). Median and interquartile range are used to express the data. Cumulative trajectory paths are presented in Figure A1.

**Figure 5 cells-11-02326-f005:**
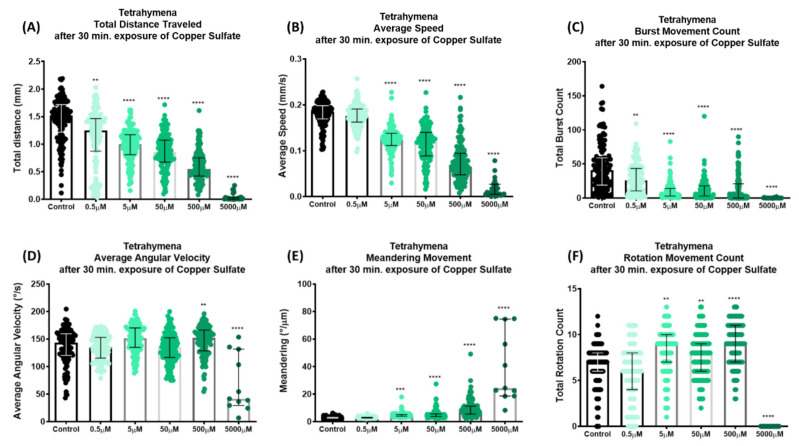
Comparison of *T. thermophila* locomotor activity after 30 min. exposure of copper sulfate. Six locomotor endpoints of (**A**) total distance traveled, (**B**) average speed, (**C**) burst, (**D**) average angular velocity, (**E**) meandering, and (**F**) rotation movement were calculated based on the 10 s video recording. The data were statistically analyzed by Kruskal–Wallis test (*n* = 129 for each group, except for 5000 μM *n* = 26; ** *p* value < 0.01; *** *p* value < 0.001; **** *p* value < 0.0001). Median and interquartile range are used to express the data. Cumulative trajectory paths are presented in Figure A1.

**Figure 6 cells-11-02326-f006:**
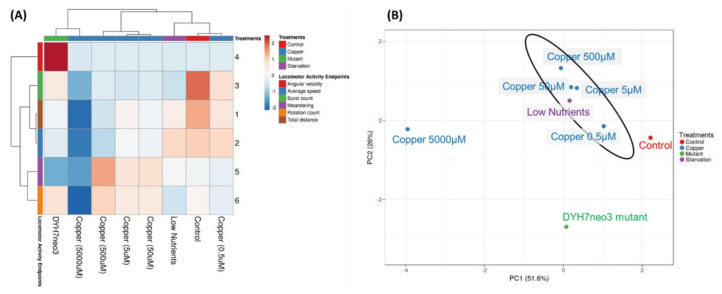
Behavior endpoint comparison in *T. thermophila* with different treatments. (**A**) Hierarchical heatmap clustering analysis and (**B**) principal component analysis (PCA). Four different treatment groups with following conditions: starvation (low nutrients) is displayed in purple color, copper exposure (0.5; 5; 50; 500; and 5000 µM) in blue color, mutant (DYH7neo3) in green color, and the untreated group is included as the control (red color).

**Table 1 cells-11-02326-t001:** Summary of all locomotor endpoints used in this study.

No.	Endpoints	Definition
1	Total distance (mm)	The total distance that the cell swam within the timeframe of the video.
2	Velocity (speed) (mm/s)	The average swimming speed of the cell.
3	Burst movement (counts)	The total amount of the cell’s movement that is higher than the upper quartile of velocity.
4	Angular velocity (°/s)	Angle of the angular speed of the cell measured in magnitude and direction.
5	Meandering (°/µm)	The degree of turning angle per travel distance.
6	Rotation movement (counts)	The amount of total clockwise and counterclockwise movement that is higher than 180°.

## Data Availability

The original data presented in this study can be obtained from the corresponding authors upon request.

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
