# Peer review of "Establishing a High-Throughput Locomotion Tracking Method for Multiple Biological Assessments in *Tetrahymena"

_cells, 2022, doi:10.3390/cells11152326_

Round 1
Reviewer 1 Report
Comments:
In this manuscript, a automated pipeline based on a deep learning-based software, TRex, for high-throughput detection of Tetrahymena motion patterns was developed. Six main locomotor endpoints, including total distance traveled, average speed and total burst movement count etc, were used to evaluate the behavioral changes of Tetrahymena under different conditions. The motion patterns of a motility mutant DYH7neo3 and Tetrahymena under low nutrition state and copper exposure were used to compare with normal Tetrahymena to verify the reliability of this method. This might provide a convenient and robust methodology for Tetrahymena locomotion tracking in high-throughput. However, there are still some problems.
Problems:
1.There may be some problems in the structure of the manuscript. The study of Tetrahymena in copper exposure is not much more prominent than the other two conditions. This method is more like a universal method for motion patterns research of Tetrahymena than a toxicity research method. Is it reasonable to take “Toxicity Assessment” as the title?
2.The manuscript mentioned that the high-throughput method regards the motion of Tetrahymena as low-dimensional, but since the motion of Tetrahymena itself is high-dimensional, what is the proportion of data errors in the overall data?
3.It is mentioned in the manuscript that this method can be measured by size. Is it that the target object cannot be distinguished from heterogeneous objects of the same size, or that all snappable objects cannot be distinguished? Can living cells and dead cells be distinguished?
4.The Tetrahymena strains used in this manuscript are Tetrahymena thermophila CU428, but what is mentioned later is Tetrahymena. It is suggested to change to Tetrahymena thermophila or T. thermophila.
Author Response
Comments and Suggestions for Authors
Comments: In this manuscript, a automated pipeline based on a deep learning-based software, TRex, for high-throughput detection of Tetrahymena motion patterns was developed. Six main locomotor endpoints, including total distance traveled, average speed and total burst movement count etc, were used to evaluate the behavioral changes of Tetrahymena under different conditions. The motion patterns of a motility mutant DYH7neo3 and Tetrahymena under low nutrition state and copper exposure were used to compare with normal Tetrahymena to verify the reliability of this method. This might provide a convenient and robust methodology for Tetrahymena locomotion tracking in high-throughput. However, there are still some problems.
Problems:
1.There may be some problems in the structure of the manuscript. The study of Tetrahymena in copper exposure is not much more prominent than the other two conditions. This method is more like a universal method for motion patterns research of Tetrahymena than a toxicity research method. Is it reasonable to take “Toxicity Assessment” as the title?
The authors appreciate the reviewer’s comment and suggestions. The authors agree with the reviewer’s point, since in this study, we did not only focus on toxicants but also on the strain evaluation and starvation condition. Thus, the title in this study is revised to ”Establish a High-Throughput Locomotion Tracking Method for Multiple Biological Assessment in Tetrahymena”. In addition, the “toxicity” words in the manuscript that referring this study as sole toxicity assessment were replaced to the term biological assessment.
2.The manuscript mentioned that the high-throughput method regards the motion of Tetrahymena as low-dimensional, but since the motion of Tetrahymena itself is high-dimensional, what is the proportion of data errors in the overall data?
Thank you for the question. In this study, we evaluated the motion of Tetrahymena by tracking their two-dimensional swimming trajectories (X and Y). The problems in TRex might occur when the identity has been lost due to individuals moving unexpectedly fast, occluded by other individuals/the environment, or simply not present in the frame. As well as what we have explained about the limitation of this study in the discussion section (line 626-628). Thus, some manual checking and correction are still needed to filter out the mistake. We estimated that there were up to 13-17% of error rate in the overall data. Additional information regarding this matter has been added to the manuscript (line 629-630).
3.It is mentioned in the manuscript that this method can be measured by size. Is it that the target object cannot be distinguished from heterogeneous objects of the same size, or that all snappable objects cannot be distinguished? Can living cells and dead cells be distinguished?
We appreciate the reviewer’s questions. It is true that TRex detect the moving animals/objects based on a range of user-defined size. So, any moving objects that are in the specified size range will be detected. This is the limitation of our study which has been mentioned in the discussion section (line 633-635). However, it’s not a major problem as long as the working environment is in clean condition. And even if there's debris or foreign matter, it's not much of a problem because usually they stay motionless. The TRex tracking ability is based on image segmentation that separating the motion objects from the background. Thus, TRex can track anything as long as the objects (or in this case, cells) move occasionally. This is also what makes living cells and dead cells distinguishable because the dead cells (motionless) will not be detected by TRex.
4.The Tetrahymena strains used in this manuscript are Tetrahymena thermophila CU428, but what is mentioned later is Tetrahymena. It is suggested to change to Tetrahymena thermophila or T. thermophila.
Thank you for your suggestion. All the words “Tetrahymena” that refer to Tetrahymena thermophila CU428 now is changed to T. thermophila to emphasize the specific species model that used in this study.
Reviewer 2 Report
The manuscript entitled “Establish a High-Throughput Locomotion Tracking Method for Toxicity Assessment in Tetrahymena” from Michael Edbert Suryanto, Ross D. Vasquez, Marri Jmelou M. Roldan, Kelvin H.-C. Chen, Jong-Chin Huang, Chung-Der Hsiao and Che-Chia Tsao describes a new method to evaluate different motility parameters of the ciliated protozoa Tetrahymena, which is proposed to be a valuable tool to study the motility of these ciliates in response to distinct environmental challenges. Indeed, Tetrahymena has been used to investigate ecotoxicity in aquatic environments. In these studies, cell viability is usually used as a criterion, but many studies have been showing that motility features of these cells, when exposed to distinct environmental conditions, may also be good markers for toxicity evaluation. For this, the authors adapted deep learning tools (software Trex) to assess multiple swimming parameters (average speed, total distance, total burst count, average angular velocity, meandering, total rotation count) in multiple cells simultaneous. Specifically, to validate their method they compared the referred parameters, determined in exponentially growing cells, with those obtained for cells subjected to stress conditions like starvation and exposure to CuSO4 both already known to affect Tetrahymena ciliate motility. Additionally, they used a mutant for a gene encoding a dynein heavy chain arm that compromises the normal swimming pattern of the cells. The obtained results were solid showing that the method allows to discriminate the differences, for a considerable high number of cells, in the different established parameters measured in control cells and in cells subjected to the different conditions studied.
The submitted manuscript is interesting and deserves to be published since it could become a useful tool, not only to study motility phenotypes in ciliates used as biological models, but also as rapid and easy tool to evaluate environmental toxicity. In the last case the authors could have investigated how the proposed method deals with complex populations of ciliates and other microorganisms expected to occur in nature. This would allow to better understand the applicability of this method to the evaluation of ecotoxicity using ciliates. It is true that they refer that the inability to change the software parameters may compromise the application of this approach to natural populations. They argue that this can be overcome using single model organisms in the laboratory exposed to the identified stresses in the wild, but still is a limitation of the method.
Minor concerns:
Line 549
“… locomotor activities were significantly reduced in the starvation condition (Fig. 4). We reason that this phenomenon might be due to a decrease in the ATP pool in Tetrahymena [55], and the lack of nutrients might also cause stress to the cell.” Starvation in Tetrahymena cells is a well characterized as a stress response and this sentence should accommodate references.
-Please correct reference 8
Author Response
Comments and Suggestions for Authors
The manuscript entitled “Establish a High-Throughput Locomotion Tracking Method for Toxicity Assessment in Tetrahymena” from Michael Edbert Suryanto, Ross D. Vasquez, Marri Jmelou M. Roldan, Kelvin H.-C. Chen, Jong-Chin Huang, Chung-Der Hsiao and Che-Chia Tsao describes a new method to evaluate different motility parameters of the ciliated protozoa Tetrahymena, which is proposed to be a valuable tool to study the motility of these ciliates in response to distinct environmental challenges. Indeed, Tetrahymena has been used to investigate ecotoxicity in aquatic environments. In these studies, cell viability is usually used as a criterion, but many studies have been showing that motility features of these cells, when exposed to distinct environmental conditions, may also be good markers for toxicity evaluation. For this, the authors adapted deep learning tools (software Trex) to assess multiple swimming parameters (average speed, total distance, total burst count, average angular velocity, meandering, total rotation count) in multiple cells simultaneous. Specifically, to validate their method they compared the referred parameters, determined in exponentially growing cells, with those obtained for cells subjected to stress conditions like starvation and exposure to CuSO4 both already known to affect Tetrahymena ciliate motility. Additionally, they used a mutant for a gene encoding a dynein heavy chain arm that compromises the normal swimming pattern of the cells. The obtained results were solid showing that the method allows to discriminate the differences, for a considerable high number of cells, in the different established parameters measured in control cells and in cells subjected to the different conditions studied.
The submitted manuscript is interesting and deserves to be published since it could become a useful tool, not only to study motility phenotypes in ciliates used as biological models, but also as rapid and easy tool to evaluate environmental toxicity. In the last case the authors could have investigated how the proposed method deals with complex populations of ciliates and other microorganisms expected to occur in nature. This would allow to better understand the applicability of this method to the evaluation of ecotoxicity using ciliates. It is true that they refer that the inability to change the software parameters may compromise the application of this approach to natural populations. They argue that this can be overcome using single model organisms in the laboratory exposed to the identified stresses in the wild, but still is a limitation of the method.
The authors highly appreciate the reviewer for taking necessary time and effort to review this manuscript. We would like to take this opportunity to thank you for the insight and expertise that you contributed towards reviewing the article, which helped us in improving the quality of this manuscript.
Minor concerns:
Line 549
“… locomotor activities were significantly reduced in the starvation condition (Fig. 4). We reason that this phenomenon might be due to a decrease in the ATP pool in Tetrahymena [55], and the lack of nutrients might also cause stress to the cell.” Starvation in Tetrahymena cells is a well characterized as a stress response and this sentence should accommodate references.
The authors thank the reviewer for pointing out this matter. A reference has been put in the sentence: “We reason that this phenomenon might be due to a decrease in the ATP pool in Tetrahymena [55], and the lack of nutrients might also cause stress to the cell [56].”
Reference 56: He, L.; Zhang, J.; Zhao, J.; Ma, N.; Kim, S.W.; Qiao, S.; Ma, X. Autophagy: the last defense against cellular nutritional stress. Advances in Nutrition 2018, 9, 493-504.
-Please correct reference 8
The authors appreciated the detailed check by the reviewer. The correction has been made for the reference 8.
Reference 8: Xiong, J.; Lu, X.; Zhou, Z.; Chang, Y.; Yuan, D.; Tian, M.; Zhou, Z.; Wang, L.; Fu, C.; Orias, E. Transcriptome analysis of the model protozoan, Tetrahymena thermophila, using deep RNA sequencing. PloS one 2012, 7, e30630.
Reviewer 3 Report
The manuscript cells-1819376 reports a very interesting methodological approach to assess the environmental toxicity using Tetrahymena thermophila as bioindicator. In particular, the exposure to various Cu concentrations has been used as pollutant condition.
The topic is important, the results are new and increase the knowledge in the field.
The manuscript is well organized and well written, the discussion is very well argued.
I consider that this manuscript is appropriate for publication in Cells, and I suggest to accept it after minor revision, including a re-reading to solve some typos in the text.
Minor revisions
The abstract exceeds the maximum number of words (200).
Page 1, lines 29-30. “Protozoa are eukaryotic unicellular microorganisms with body size of the micrometer scale.”. Being the protozoa of microorganisms, it is obvious that they have body size of the micrometer scale.
Page 5, lines 217-219. “A total of 100 μL CuSO4 were immersed by mixing it with the cells on the slide until reach the desired working concentrations (0.5, 5, 50, 500, and 5000 μM)”. Authors should justify their choice to use this metal and this concentration range. For example, there are some studies, even recent ones, in which the Cu was evaluate because an oxidative stress inducer, and the concentration of 500 μM was used as the maximum that does not cause toxicity for the Tetrahymena cells, based on the growth curves. It would be advisable for the authors to cite some of these papers.
Author Response
Comments and Suggestions for Authors
The manuscript cells-1819376 reports a very interesting methodological approach to assess the environmental toxicity using Tetrahymena thermophila as bioindicator. In particular, the exposure to various Cu concentrations has been used as pollutant condition.
The topic is important, the results are new and increase the knowledge in the field. The manuscript is well organized and well written, the discussion is very well argued. I consider that this manuscript is appropriate for publication in Cells, and I suggest to accept it after minor revision, including a re-reading to solve some typos in the text.
The authors thank the reviewer for the careful and insightful review of our manuscript. We sincerely appreciate all your valuable comments and suggestions, which helped us in improving the quality of the manuscript. Careful check and editing have been made to correct some language mistakes in the manuscript.
Minor revisions
The abstract exceeds the maximum number of words (200).
We appreciate the reviewer’s suggestion. Previously the number of words in abstract was 224, and now it has been reduced to 209. The authors had tried their best to make the abstract concise enough without neglecting the necessary information of this research. Hopefully, it will be acceptable to you.
Page 1, lines 29-30. “Protozoa are eukaryotic unicellular microorganisms with body size of the micrometer scale.”. Being the protozoa of microorganisms, it is obvious that they have body size of the micrometer scale.
Thank you for the valuable comment. The authors also agree with the reviewer and found this sentence was problematic and contained repetition of meaning. Thus, the sentence “with body size of the micrometer scale” was deleted. The sentence has been revised to “Protozoa are eukaryotic unicellular microorganisms that have important ecological role, easy handling, and rapid growth which are suitable for ecotoxicity assessment”.
Page 5, lines 217-219. “A total of 100 μL CuSO4 were immersed by mixing it with the cells on the slide until reach the desired working concentrations (0.5, 5, 50, 500, and 5000 μM)”. Authors should justify their choice to use this metal and this concentration range. For example, there are some studies, even recent ones, in which the Cu was evaluate because an oxidative stress inducer, and the concentration of 500 μM was used as the maximum that does not cause toxicity for the Tetrahymena cells, based on the growth curves. It would be advisable for the authors to cite some of these papers.
The authors appreciated the reviewer for raising this important point. The reason of choosing the CuSO4 (line 213-215) and the working concentrations (line 222-223) have been added to the manuscript. CuSO4 was selected as the representative of toxicant because of its heavy metal characteristic that well known has profound effects on ciliated protozoans (Rico et al., 2009) (Ref 43). And as the reviewer suggested, we put citation to justify the implementation of the selected working concentrations, which is based on the previous study that evaluate the CuSO4 concentrations on the growth rate of Tetrahymena cells (Ferro et al., 2020) (Ref 44).
Reference 43: Rico, D., Martín-González, A., Díaz, S., de Lucas, P., & Gutiérrez, J. C. (2009). Heavy metals generate reactive oxygen species in terrestrial and aquatic ciliated protozoa. Comparative Biochemistry and Physiology Part C: Toxicology & Pharmacology, 149(1), 90-96.
Reference 44: Ferro, D., Bakiu, R., Pucciarelli, S., Miceli, C., Vallesi, A., Irato, P., & Santovito, G. (2020). Molecular characterization, protein–protein interaction network, and evolution of four glutathione peroxidases from Tetrahymena thermophila. Antioxidants, 9(10), 949.